# Removal of Heavy Metal Ions from Wastewater Using Hydroxyethyl Methacrylate-Modified Cellulose Nanofibers: Kinetic, Equilibrium, and Thermodynamic Analysis

**DOI:** 10.3390/ijerph18126581

**Published:** 2021-06-18

**Authors:** Mohamed Gouda, Abdullah Aljaafari

**Affiliations:** 1Department of Chemistry, College of Science, King Faisal University, Al-Ahsa 31982, Saudi Arabia; 2Department of Physics, College of Science, King Faisal University, Al-Ahsa 31982, Saudi Arabia; aaljaafari@kfu.edu.sa

**Keywords:** microwave technique, electrospinning, grafting copolymerization, cellulose nanofiber, heavy metal sorption

## Abstract

The objective of this work was to fabricate modified cellulose nanofibers (CNFs) for the removal of heavy metal ions (Cd^2+^ and Pb^2+^) from wastewater. Cellulose was modified with 2-hydroxyethyl methacrylate (HEMA) via grafting copolymerization using the microwave-assisted technique in the presence of ceric ammonium nitrate (CAN) as an initiator. Prepared cellulose-graft-(2-hydroxyethyl methacrylate) (HEMA/C) copolymers were characterized using Fourier transform infrared spectroscopy (FT-IR) and scanning electron microscopy (SEM). Different factors affecting the graft yield, such as irradiation time, monomer concentrations, and initiator concentrations, were investigated. Furthermore, cellulose-graft-(2-hydroxyethyl methacrylate) copolymer nanofibers (HEMA/CNF) were fabricated by electrospinning using *N*,*N*-dimethylacetamide-LiCl as a solvent. Electrospun nanofiber copolymers were characterized using SEM and thermogravimetric analysis (TGA). Operating parameters, including time, starting metal concentrations, and adsorbent concentration, were analyzed at a pH of 5.6 for the two metal ions. The best-fit model of adsorption energy was the pseudo-second-order model, and adsorption isotherms at equilibrium were well described by the Langmuir and Freundlich models. The negative values of ΔG and positive values of ΔH and ΔS suggest that the adsorption of Cd^2+^ and Pb^2+^ ions onto electrospun HEMA/CNF is a spontaneous, endothermic, and favorable reaction.

## 1. Introduction

Freshwater requirements continue to rise worldwide as a result of population growth, urbanization, food and energy security demands, global economic processes such as international trade, changing diets, expanding agribusiness, and atmosphere changes [1]. Global water demand is anticipated to increase by 55% from 2015 to 2050, primarily because of increasing industry and household use [1]. At the same time, water contamination has become a crucial issue throughout the world. A wide range of wastewater sources, such as household and industrial effluents, sewage, stormwater, leachate, and septic tank wastewater, are viewed as wastewaters by society [2]. Industrial wastewater is a significant source of contamination in water bodies. Generally, industrial wastewater contaminants can be divided into two groups: inorganic and organic compounds [3]. Industrial waste, untreated sewage, oil slicks, harmful synthetics, radioactive wastes, and inorganic compounds, such as heavy metals that are released from processing plants and treatment facilities, all contribute to the deterioration of water quality [4]. Consequently, unless demand decreases or supply increases, the world will experience progressively extreme water deficiencies.

Heavy metals are typically elements with atomic weights in the range 63.5–200.6 [5]. Although every life form requires some metals to survive, such as cobalt, copper, iron, and manganese, excessive heavy metals discharged in enormous amounts, for example, cadmium, chromium, mercury, lead, arsenic, and antimony, are of incredible ecological concern [5]. Exposure to such metals can cause severe negative health effects, such as decreased development and growth, malignancy, organ toxicity, sensory system harm, and, in extraordinary cases, death [6]. Many heavy metal contaminants are present in wastewaters that are released by industries involving metallurgical/metal assembling, electroplating, and mining; printing, coloring, and painting; petrochemical activities; and the production of pulp and paper, textiles, and batteries and chemicals [6,7,8]. These industries directly or indirectly release large amounts of heavy metals into the environment, where they can accumulate in deadly amounts, especially in developing countries that lack appropriate waste treatment protocols [4,7,8]. Numerous standard wastewater treatment procedures, such as chemical precipitation, adsorption, particle exchange, and electrochemical methods, have been utilized to remove and recover heavy metals from contaminated waters [6]. Additionally, more current methodologies, including layer partition and electrodialysis, have been created to increase the extent of metal removal from wastewaters. Membrane-based methods, such as reverse osmosis, nanofiltration, ultrafiltration, and microfiltration, are generally utilized for heavy metal removal from contaminated water [9].

Compared with other traditional strategies, resin-based ion-exchange media and macroporous ion-exchange membrane layers allow a much higher volumetric throughput [8]. However, their cost-efficiency for metal recovery depends on the sorption capacity, which is strongly related to the surface area, and particle selectivity, which depends on the membrane material. Therefore, the development of new, improved membranes with higher surface area, high particle selectivity, and high volumetric throughput is a growing study area in the water purification field.

Basic polymeric membrane manufacturing strategies, such as stage reversal, interfacial polymerization, stretching, track-etching, and electrospinning, have been utilized to fabricate polymeric membranes [10]. The choice of the procedure for polymer layer manufacturing depends on the polymer and the ideal structure of the film.

Electrospinning can be utilized to create nanofiber films with large surface-area-to-volume ratios and high sorption capacities to remove contaminants from natural sources [11,12]. Additionally, adsorptive nanofiber films have high porosities and interconnected pore structures that enable higher efficiency compared with conventional barrier layer methods, for example, turnaround assimilation and nanofiltration [13].

Cellulose is one of the numerous polymers that can be utilized to fabricate nanofiber films. Cellulose is a polysaccharide comprising a straight chain of a few hundred to more than 10,000 β-(1→4)-linked D-glucose units with the molecular formula (C_6_H_10_O_5_)_n_ [14]. As a carbohydrate polymer, the atomic structure of cellulose has countless hydroxyl groups (three for every anhydroglucose unit), which involve extensive intra- and intermolecular hydrogen bonds. Cellulose is unscented, non-poisonous, and biodegradable. It is ubiquitous in nature and incorporated into products used daily by individuals, such as paper and textiles [15].

Wastewater contaminated with poisonous metals has been treated with a variety of materials, including nanocomposites [16], nanofilters [17], synthetic nanomaterials [18], chelating minerals [19], activated carbon [20], and biopolymers [21].

Numerous factors can increase the effectiveness of nanofibrous membranes made of polymers (for example, cellulose and chitosan) for environmental remediation: high surface capacity, small interfiber spacing, and high gas permeability can increase the capacity of adsorption/chelation for remediation. Nanomembranes made of polymers can have extraordinary adsorption selectivity and capacity because of their functional moieties, for example, SO_3_H, COOH, and NH_2_, through which the metal contaminants are chelated. Researchers have studied the removal of harmful heavy metals using biopolymer-based adsorbents, for instance, cellulose. Abou-Zeid et al. [22] employed tetramethylpiperidine oxide oxidation with and without polyamide-amine-epichlorohydrin crosslinker to form 2,3,6-tricarboxy cellulose nanofiber as a bio-adsorbent, which they used for the treatment of water containing calcium, lead, and copper particles. In another study, phosphoric acid was used to functionalize cellulose nanofibers with a phosphate group for the removal of copper particles from water [23]. Recently, Choi et al. fabricated cellulose nanofibers by deacetylating electrospun cellulose acetic acid nanofibers, followed by their functionalization with a thiol group introduced through an esterification reaction. This polymer-based adsorbent was used for the removal of Pb (II), Cd (II), and Cu (II) particles. Metal particles were removed by means of a complexation reaction between the two surface thiol groups and divalent metals to confirm the effectiveness of the functionalized nanomaterial for water remediation [24]. In addition, the previous methodology was used for the fabrication of cellulose nanofibrous mats that were then modified using citric acid. The citric acid-modified cellulose nanofibers were developed for the removal of chromium particles through a cluster adsorption test. This fused nano-adsorbent was highly effective in removing Cr (VI) from an aqueous solution [25]. 

The aim of this work was to fabricate nanofiber membranes based on cellulose-graft-poly(2-hydroxy-ethylmethacrylate) copolymers (HEMA/C) and used them to eliminate cadmium and lead ions (Cd^2+^ and Pb^2+^) from ionic solutions. The membranes were synthesized using a microwave radiation (0–1800 W)-assisted technique in the presence of ceric ammonium nitrate as an initiator of free radical formation. The presence of poly-HEMA on the cellulose backbone was characterized by scanning electron microscopy and FTIR spectroscopy. HEMA/CNF was fabricated using an electrospinning method. The surface morphology and thermal stability of electrospun HEMA/CNF copolymers were characterized by scanning electron microscopy and TGA, respectively. The performance of electrospun HEMA/CNF copolymers in removing heavy metals from aqueous ionic solutions was investigated. The effects of various adsorption process parameters, including solution pH, adsorption time, starting concentration of heavy metal ions, and dose of adsorbent, were analyzed. Furthermore, the adsorption kinetics, isotherms, and thermodynamics were examined. 

## 2. Materials and Methods

### 2.1. Materials

Cellulose with a molecular weight of 30,000, 2-hydroxyethyl methacrylate (HEMA), ceric ammonium nitrate (CAN) as an initiator, *N*,*N*-dimethylacetamide, and lithium chloride were purchased from Sigma-Aldrich, St. Louis, MO, United States. Cadmium chloride and lead chloride were purchased from Merck Co., Darmstadt, Germany. Phosphate buffer solutions were purchased from Merck Co., Darmstadt, Germany. All chemicals were used as received without any additional purification. 

### 2.2. Synthesis of HEMA/C Copolymer 

HEMA was grafted onto cellulose using a microwave irradiation method with a microwave operating system (CEM 6 Matthews, VA United States). The operating power varied from 0 to 1800 W, and the frequency was 2455 MHz. The specified number of moles (M/L) of CAN was added to an Erlenmeyer glass beaker containing 50 mL of distilled water and 1 g of cellulose powder. The specified HEMA monomer concentration (M/L) was added to the above mixture with continuous stirring. The mixed solution was removed from the microwave vessel and then placed on the turntable of the microwave. The mixed solution was subjected to microwave irradiation (1000 W) for the specified time until a gel was formed. Then, the mixed solution was cooled by the cooling fan of the microwave system. This microwave irradiation/cooling cycle was repeated until a bulk gel was produced. Finally, the formed bulk gel was transferred to a beaker containing acetone to separate the grafted copolymer from impurities such as any remaining initiator, ungrafted monomer, and formed homopolymers. 

### 2.3. Purification of Prepared HEMA/C Copolymers 

HEMA/C copolymers were isolated from the remaining initiator, *p-HEMA* homopolymers, and ungrafted *HEMA* monomers by solvent extraction. The cellulose graft copolymers were soaked in methanol using Soxhlet extraction until the graft copolymer reached a constant mass. HEMA/C copolymers were dried in an oven at 60 °C for 2 h. Grafting yield percentage (*Gy*%) was calculated by the following Equation (1):(1)$$Gy\%=\frac{Weight\;of\;cellulose-g-p-HEMA\;-\;Weight\;of\;Cellulose}{Weight\;of\;cellulose}\;\times \;100.$$

### 2.4. Synthesis of HEMA/CNF 

A solution of HEMA/C copolymer was prepared as follows: 10 g of HEMA/C copolymer was added to 100 mL of *N*,*N*-dimethylacetamide, and the mixture was transferred to a thermoplate adjusted to 130 °C for 2 h with constant stirring until the polymer was dissolved. After the slurry was cooled to 100 °C, anhydrous lithium chloride (15 g) was added. Then, the mixture was cooled to room temperature with stirring until the HEMA/C copolymers were completely liquefied. The prepared solutions were transferred to an electrospinning unit (NEU-010, Kato Tech CO., LTD, Kyoto, Japan). In the electrospinning process, a high electric field was applied to a drop of HEMA/C copolymer solution at the tip of a nozzle (internal width 0.20 mm). A high voltage (25 kV) was applied to the collecting target drum by a high-voltage power source. The flow rate of the polymer solution was 1 mL/h and driven by compressive force. The electrospun HEMA/CNF copolymers were collected on the target drum located at a 15 cm distance from the needle tip. The spinning rate of the rotating target drum was maintained at 45 ± 5 r/min. The electrospinning method was performed in laboratory conditions (i.e., temperature = 25 ± 1 °C; relative humidity = 71 ± 3%). 

### 2.5. Characterization

Fourier-transform infrared spectroscopy (FTIR) examination was applied to confirm the attachment of HEMA onto the cellulose structure using an FTIR spectrometer (Shimadzu Scientific Instrument FT-IR-8400S, Columbia, MD, USA), while scanning electron microscopy (JEOL JXA-840-An electron test microanalyzer-scanning electron magnifying lens, JEOL ltd., Tokyo, Japan) was utilized to investigate the surface morphology of both HEMA/C copolymers and electrospun HEMA/CNF. Then, thermogravimetric analysis (TGA) was performed on a thermogravimetric analyzer (TA Instruments, model Q 500-, New Castle, DE, USA) to quantify the thermal stability of HEMA/C copolymers and electrospun HEMA/CNF. The analysis was carried out using 1–2 mg samples in a dynamic nitrogen environment with a heating rate of 10 °C/min from room temperature to 650 °C.

### 2.6. Removal of Heavy Metal Ions 

Electrospun HEMA/CNF copolymer nanofibers were used to remove Cd^2+^ and Pb^2+^ ions from aqueous ionic solutions. Electrospun HEMA/CNF copolymer (0.1 g) was immersed in 20 mL of Cd^2+^ and Pb^2+^ ion solutions with different concentrations (mg/L) at various pH values (3–7). Prepared solutions were stirred for different contact times (15–150 min). Atomic absorption spectroscopy was performed on a VarianSpectrAA-220 system (Varian, Australia Pty Ltd., Belrose, Australia) equipped with a graphite heater and deuterium arc background corrector as indicated by APHA 2005 [26] to determine the Pb^2+^ and Cd^2+^ concentrations before and after contact with electrospun copolymer nanofibers. The amount of metal adsorbed onto the adsorbent was calculated according to Equations (2) and (3):(2)$$q_{t}=\frac{\left(C_{0}-C_{t}\right)}{m}V,$$
(3)$$q_{c}=\frac{\left(C_{0}-C_{c}\right)}{m}V,$$
where *q_t_* is the adsorbed metal content at time t, and qe is the adsorbed metal content at equilibrium (mg g^−1^). *C*_0_, *C_t_*, and *C_e_* (ppm = mg L^−1^) are the concentrations of metal at the starting time, time t, and equilibrium, respectively; *V* (L) is the metal solution volume, and *m* (g) is the weight of the electrospun nanofiber. The removal efficiency (*R*%) was determined according to the Equation (4):(4)$$R\%=\frac{\left(C_{0}-C_{t}\right)}{C_{0}}\times 100.$$

Analysis of the adsorption kinetics is essential, as it reveals the rate at which the adsorbate is taken up by the adsorbent. The mechanism and rate of adsorption on the electrospun nanofibers can be determined based on kinetic analysis. To reveal the adsorption kinetics, pseudo-first-order and pseudo-second-order models were utilized according to Equations (5) and (6).
(5)$$q_{t}=q_{e}\left[1-exp\left(-k_{1}t\right)\right],$$
(6)$$q_{t}=\frac{k_{2}q_{e}{}^{2}t}{1+k_{2}q_{e}t},$$
where *k*_1_ and *k*_2_ are the constant rates of adsorption in pseudo-first-order and pseudo-second-order reactions, respectively. The pseudo-first-order kinetic model tends to be more appropriate for low concentrations of metal ions and is described by the following Equation (7): (7)$$\ln\left(q_{e}-q_{t}\right)=\ln q_{e}-k_{1}t,$$
where *q_e_* is the metal content adsorbed at equilibrium per gram of electrospun nanofiber (mg g^−1^), *q_t_* is the metal content adsorbed at time t per gram of electrospun nanofiber (mg g^−1^), and *k*_1_ (min^−1^) is the pseudo-first-order adsorption rate constant. In contrast, the pseudo-second-order kinetic model is dependent on the quantity of solute adsorbed on the surface of electrospun nanofibers and the adsorbed quantity at equilibrium.

#### 2.6.1. Adsorption Isotherms

The capacity of electrospun nanofiber adsorption at different initial concentrations of metal ions at equilibrium can be determined by the isotherms of the adsorption. The isotherms of adsorption indicate how the adsorbate interacts with the adsorbent and provide a thorough understanding of their relationship. Several isotherm conditions were analyzed for this investigation, and the two significant isotherms were utilized.

1. Monolayer adsorption on a surface containing a limited number of binding sites, assuming insignificant interactions among adsorbed ions and constant adsorption energy, was represented by Langmuir’s isotherm. The maximum adsorption depends on the capacity of the monolayer surface. The Langmuir isotherm is characterized by the following linear Equation (8):(8)$$\frac{q_{c}}{C_{c}}=\frac{1}{K_{L}q_{m}}+\frac{C_{c}}{q_{m}},$$
where *q_e_* is the concentration at equilibrium, *C_e_* is the liquid-phase concentration in mole L^−1^, *q_m_* is the maximum adsorption capacity of the monolayer (mg g^−1^), and *K_L_* is the constant at equilibrium (L mol^−1^). A straight line with a slope of 1/*q_m_* and an intercept of 1/*q_m_K_L_* is acquired when *C_e_*/*q_e_* is plotted against *C_e_*. The separation factor (*R_L_*) is a dimensionless constant, which is a basic attribute of the Langmuir model. The formula for *R_L_* is calculated as according to Equation (9):(9)$$R_{L}=\frac{1}{\left(1+K_{L}C_{o}\right)},$$
where *C_o_* (mg L^−1^) is the starting concentration of the adsorbate (*C_o_* = 150 mg L^−1^). The isotherm is unfavorable if *R_L_* > 1, linear if *R_L_* = 1, favorable if 0 < *R_L_* < 1, and irreversible if *R_L_* = 0.

2. Adsorbents that follow the Freundlich isotherm model are expected to have a heterogeneous surface comprising different types of adsorption sites, and a monolayer is assumed for ion adsorptions, as in the Langmuir condition as the following Equation (10): (10)$$\ln q_{c}=\ln K_{f}+\frac{1}{\left(n\ln C_{c}\right)},$$
where *K_f_* is a constant (related to the adsorption energy), and *n* is a constant that defines the intensity of adsorption; a plot of ln *q_e_* versus ln *C_e_* provides a straight line with a slope of 1/*n* and intersection of ln *K_f_*. The value of “*n*” indicates the adsorption favorability.

#### 2.6.2. Adsorption Thermodynamics

Adsorption thermodynamics were analyzed with an initial concentration of 100 mg L^−1^ adsorbed ions at 25 and 35 °C. The thermodynamic parameters were determined by the following equations:(11)$$K_{D}=q_{e}/C_{e},$$
(12)$$\Delta G=-RT\;\ln K_{D},$$
(13)$$\ln K_{D}=\left(\Delta S/R\right)-\left(\Delta H/RT\right),$$
where *K_D_* is the dissociation constant at equilibrium, Δ*G* (kJ/mol) is the Gibbs free energy change, *R* (8.314 J/mol K) is the ideal gas constant, *T* (K) is the temperature, Δ*H* (kJ/mol) is the enthalpy change, and Δ*S* (kJ/mol K) is the entropy change. The values of Δ*G* were determined from the *K_D_* values for every temperature, and the values of Δ*H* and Δ*S* were determined from the slope and intercept of the plot of ln *K_D_* versus 1/*T*, respectively.

## 3. Results and Discussion

### 3.1. Synthesis of HEMA/C Copolymers

HEMA/C copolymers were prepared using a microwave-assisted technique. Different parameters affected the grafting of HEMA onto the cellulose surface, such as the concentration of ceric ammonium nitrate (CAN), microwave irradiation time, and HEMA concentrations. Figure 1 indicates that the grafting yield of HEMA was best with a microwave irradiation time of 80 s, a HEMA concentration of 15 M/L, and a CAN initiator concentration of 0.3 M/L. Microwave radiation assists in the production of only polar bonds, which facilitate the generation and reaction of free radicals. CAN was applied to oxidize the primary hydroxyl group of cellulose by withdrawing the electron from the oxygen of primary hydroxyl in cellulose and forming a new bond (Cell-O-CAN), which is more polar than Cell-OH and easy to break down with microwave radiation to create free radical sites on the backbone of cellulose [27]. When using a CAN concentration of more than 0.3 M/L, the graft yield % decreased due to the termination reaction via the coupling of initiator free radicals. The second step was chain propagation, in which the addition reaction occurred between the generated cellulose free radicals and the monomer. Using monomer concentrations of more than 15 M/L decreased the graft yield % due to the termination reaction occurring by either a coupling reaction between monomer free radicals or a disproportionation reaction. Furthermore, the decrease in grafting yield % may be due to the degradation of CAN at an irradiation time longer than 80 s. Scheme 1 shows the suggested reaction mechanism of grafting HEMA onto cellulose using the microwave-assisted technique.

### 3.2. Characterization of HEMA/C Copolymer 

#### 3.2.1. FTIR

Figure 2 shows the FTIR spectra of native cellulose and HEMA/C copolymer. The FTIR spectra show C-H and C=O stretching bands of methyl and methylene groups at 2900 and 1710 cm^−1^, respectively. Furthermore, there is a characteristic band at 750 cm^−1^ for -OH bending of p-HEMA in graft copolymers. The obtained data from FTIR spectra confirm the existence of poly-HEMA on the backbone of cellulose.

#### 3.2.2. Scanning Electron Microscopy

The surface morphology of native cellulose and cellulose-graft-copolymers was characterized by scanning electron microscopy, as shown in Figure 3a,b. SEM images reveal that the surface of the cellulose was unchanged through the grafting process, as bonds of the graft copolymers were formed in both the amorphous and crystalline areas.

### 3.3. Characterization of Electrospun HEMA/CNF Copolymer Nanofibers

#### 3.3.1. SEM

Figure 4 provides SEM pictures showing the surface morphology of native cellulose nanofibers and HEMA/CNF copolymers. The electrospun nanofibers (Figure 4A,B) have widths ranging from 200 to 300 nm. The grafting process slightly affected the surface morphology of the HEMA/CNF copolymer; HEMA/CNFs were slightly bent and fused together, as previously observed [28]. This outcome can be explained by hydrogen bonding among the hydroxyl groups of HEMA/CNF. Rough surfaces were observed after grafting with pHEMA, as demonstrated in Figure 4B.

#### 3.3.2. TGA

The weight loss of electrospun HEMA/CNF copolymers during heating was determined using thermogravimetric analysis. The thermal behavior of the nanofibers was monitored up to a temperature of 650 °C. The TG curves for electrospun native cellulose nanofibers or electrospun HEMA/CNF copolymers show rapid weight loss until about 350 °C and 430 °C, respectively, after which no additional weight loss occurred, as shown in Figure 5. The thermal stability of electrospun HEMA/CNF copolymers is a little higher than that of electrospun native cellulose nanofibers. The increase in the stability of thermal stability of HEMA/CNF may be due to the integration of more covalent bonds between the poly-HEMA chains and cellulosic fibers. The grafted molecule forms a crosslinked network with the cellulosic filaments, which, when heated, serves as a protective energy barrier on the surface, thus resisting degradation and consequently increasing the thermal stability of bonded cellulose chains [29].

### 3.4. Sorption of Heavy Metal Ions

#### 3.4.1. Effect of pH Values

Figure 6 shows the effect of different pH values (3–7) on the adsorption percentage of Cd^2+^ and Pb^2+^ using heavy metal concentrations of 50 mg/L and an adsorption contact time of 1 h. The data reveal that the adsorption of heavy metal ions increased as the pH value increased from 3 to 5, after which it decreased. This is attributed to the fact that, at low pH, the nanofiber surface is surrounded by hydronium ions (H_3_O^+^) and has mostly protonated sites, resulting in a net positive charge on the nanofiber surface. This positive charge prevents the metal ions from contacting the surface of functionalized nanofibers [30]. Metal ion adsorption from the aqueous ionic solution decreased in the following order: Pb^2+^ < Cd^2+^. This is in accordance with the William-Irving series.

#### 3.4.2. Effect of Contact Time

The data from Figure 7 show that the adsorption percentage of Cd^2+^ and Pb^2+^ increased as the contact time of nanofibers with the metal ions increased for all contact times tested, and the maximum adsorption occurred at 160 min, after which the adsorption % slightly decreased. This may be attributed to a decrease in the number of adsorbent sites and a slight decrease in the metal ion concentration in the solution.

#### 3.4.3. Effect of the Concentration of Metal Ions 

As shown in Figure 8, the adsorption percentage of heavy metal ions from the aqueous ionic solution depended on the starting concentration of the metals. The data reveal that the adsorption percentage decreased as the concentration of heavy metals increased. This is due to the increase in the driving force from the gradient concentration.

#### 3.4.4. Adsorption Kinetics

The pseudo-first-order and the pseudo-second-order kinetic models were used to test the effect of changing the adsorption time on the adsorption rate, as shown in Figure 9. These models were useful in understanding the adsorption process of Cd^2+^ and Pb^2+^ at a starting concentration of 50 mg L^−1^ using the prepared electrospun HEMA/CNF copolymer adsorbent. The kinetic rate constants were determined using conditions (5), (6), and (7) in the experimental section. 

The pseudo-first-order model assumes that the adsorption limit is confined to a single system with one type of adsorption site [31]. By comparing the experimental adsorption results with known values, it was discovered that the measured values were far from their reference values. Consequently, the pseudo-first-order response kinetic model was not suitable for describing the adsorption of Cd^2+^ and Pb^2+^ onto electrospun HEMA/CNF copolymers. 

The pseudo-second-order kinetic model is dependent on the amount of adsorbate adsorbed on the adsorbent surface and the amount adsorbed at equilibrium. As outlined in Table 1, the known *q_e_* values agree with the test *q_e_* values (*q*_*e*,*exp*_), and the correlation coefficient (*R*^2^) values are consistently higher than those of the pseudo-first-order model. Thus, the adsorption data strongly agree with the pseudo-second-order model, which implies that the rate-controlling step is chemisorption, and the adsorption rate for the two ions depends on the availability of adsorption sites on the surface of adsorbent materials. Additionally, as the starting concentrations of Cd^2+^ and Pb^2+^ increased, the *k*_2_ values decreased, which might be attributable to increased competition for the adsorption sites at high concentrations compared with low concentrations [32].

#### 3.4.5. Adsorption Isotherms

By utilizing Langmuir and Freundlich adsorption isotherm models [33,34] at various starting concentrations and two different temperatures (25 °C and 35 °C), as shown in Table 2, it can be inferred that the Langmuir model best characterizes the adsorption data, and the adsorption process is more effective at high temperatures. Additionally, for the Freundlich model, the values of 1/*n_F_*, which measure the adsorption strength or surface heterogeneity, change in the range 0–1, which indicates greater heterogeneity as the value approaches zero. The results also demonstrate that the adsorption of the two metals is well described by the Langmuir and Freundlich isotherm models.

#### 3.4.6. Adsorption Thermodynamic 

Thermodynamic parameters, i.e., Gibbs free energy (Δ*G*), enthalpy (Δ*H*), and entropy (Δ*S*), were investigated at 25 and 35 °C, as shown in Equations (8)–(10) and reported in Table 3. The negative values of Δ*G* indicate that Cd^2+^ and Pb^2+^ adsorbent reactions are spontaneous [35], and this characteristic increases as the temperature increases. The positive values of Δ*H* demonstrate the endothermic nature of the adsorption reaction, while the positive values of Δ*S* indicate an increase in entropy at the solid–liquid interface during the adsorption reaction [36]. In sum, it is concluded that the adsorption of Cd^2+^ and Pb^2+^ on the HEMA/CNF copolymer is an endothermic, favorable, and spontaneous reaction. 

## 4. Conclusions

Cellulose-graft-poly(2-hydroxyethyl methacrylate) copolymers were synthesized using a microwave-assisted method. The optimum grafting yield was obtained at a microwave irradiation time of 80 s, monomer concentration of 15 M/L, and initiator concentration of 0.3 M/L. The synthesized graft copolymer was characterized using FTIR spectroscopy to confirm the presence of the polymer on the cellulose backbone. Furthermore, scanning electron microscopy was used to study the surface morphology of the grafted cellulose. Electrospun cellulose-graft-poly(2-hydroxyethyl methacrylate) copolymer nanofibers were synthesized using the electrospinning method. Electrospun graft copolymer nanofibers were characterized using scanning electron microscopy, and the result shows that the electrospun nanofibers had diameters in the range 200–300 nm. In addition, the grafted cellulose nanofibers were slightly curved and fused together. Thermogravimetric analysis (TGA) was used to determine the thermal stability of electrospun grafted copolymer nanofibers, and the data reveal that the grafted electrospun grafted copolymer nanofibers were more stable than the ungrafted electrospun nanofibers. The potential of the electrospun grafted copolymer nanofibers to remove heavy metals from aqueous ionic solutions was investigated. The effects of different parameters on the absorption percentage of heavy metals, such as initial metal concentrations, contact time, and pH, were studied. The obtained results show that the optimum absorption percentage was at a pH of 5, an initial concentration of 30 mg/L, and a contact time of 180 min. The test results indicate that the adsorption equilibrium for electrospun HEMA/C copolymer nanofibers was reached in just 13.3 min for 50 mg L^−1^ Cd^2^ and in 3.0 min for 100.5 mg L^−1^ Pb^2+^. In summary, the prepared electrospun HEMA/C copolymer nanofibers showed effective adsorption of Cd^2+^ and Pb^2+^ at a high environmental concentration as well as a low concentration. 

## Data Availability

Not applicable.

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
