# Peer review of "Removal of Heavy Metal Ions from Wastewater Using Hydroxyethyl Methacrylate-Modified Cellulose Nanofibers: Kinetic, Equilibrium, and Thermodynamic Analysis"

_ijerph, 2021, doi:10.3390/ijerph18126581_

Round 1

Reviewer 1 Report

Remarks

1). Equation (6) should be written correctly

2). Why was the kinetic study not conducted to equilibrium?

3). The unit [k2 (min-1)] of the pseudo-second order kinetic constant given in Table 1 is incorrect

4). The standard deviation error bars should be added in Figures 6, 7, 8. 

Author Response

1). Equation (6) should be written correctly

 Response 2: Corrected

2). Why was the kinetic study not conducted to equilibrium?

Response 2: the Authors have introduced the thanks to the reviewer for his comment and they revised and conduct the kinetic study with the equilibrium time

3). The unit [k2 (min-1)] of the pseudo-second-order kinetic constant given in Table 1 is incorrect

Response 3: The Authors are grateful for the reviewer's notice but unfortunately the Authors found the unit is correct (min-1) it is like the pseudo-first-order.

4). The standard deviation error bars should be added in Figures 6, 7, 8. 

Response 4: Thanks a lot but the Authors cannot add the required comments at this time due to all samples are exhausted and in the next work we will put in our consideration this notice.

Reviewer 2 Report

There are hundreds of scientific publications showing the performance of many natural or synthetic materials in adsorbing metals in solution. From this point of view, this manuscript is not particularly original. The preparation and use of an adsorbent based on modified cellulose nanofibers is however of some interest, at least from a scientific point of view.

The manuscript is well written and easy to follow. The experimental conditions used to test the adsorbent are fairly standard. However, some information is missing:

1) What types of salts (nitrates, chloride, etc.) were used for the preparation of the metal solutions?
2) What alkaline product was used to adjust the pH?
3) Figure 6: Were the tests done in triplicate? If so, add the error bars.
4) Is it possible to elute the metals and reuse the adsorbent for several adsorption-elution cycles?
5) How does the performance of the adsorbent compare to other adsorbents tested in the literature for adsorption of Cd and Pb under comparable conditions?
6) The method of preparing the adsorbent seems to complicate for an environmental application which should not be expensive. Could the authors discuss the feasibility of producing this adsorbent on a large scale?
7) Were controls used to verify that metal removal was not also attributable to precipitation of metals in the form of metal hydroxides?

Author Response

1) What types of salts (nitrates, chloride, etc.) were used for the preparation of the metal solutions?

Response 1: Cadmium chloride and lead chloride were used

2) What alkaline product was used to adjust the pH?

Response 2: Phosphate buffer solutions

3) Figure 6: Were the tests done in triplicate? If so, add the error bars.

Response 3: Yes, and the data that plotted are the average of the triplicate tests

4) Is it possible to elute the metals and reuse the adsorbent for several adsorption-elution cycles?

Response 4: Yes, it is possible to reactivate the adsorbent materials, and this can be done in the future due to the exhaustion of the prepared samples

5) How does the performance of the adsorbent compare to other adsorbents tested in the literature for adsorption of Cd and Pb under comparable conditions?

Response 5:

6) The method of preparing the adsorbent seems to complicate for an environmental application which should not be expensive. Could the authors discuss the feasibility of producing this adsorbent on a large scale?

Response 6: We are currently in the process of studying the feasibility of producing this type of nanofibre

7) Were controls used to verify that metal removal was not also attributable to precipitation of metals in the form of metal hydroxides?

Response 7: Yes, this was confirmed by studying the effect of the pH, where care was taken to study the adsorption on transparent solutions.

Reviewer 3 Report

The paper by Gouda and Aljaafari (ijerph-1248383) deals with nanofiber synthesis and adsorption of heavy metals. Nanofiber adsorbents is an important trend in adsorption field but this manuscript is very elementary and does not provide any new insights. Moreover, the quality is unacceptably low. I do not recommend publication in International Journal of Environmental Research and Public Health.

General comments:

  1. The language must be re-checked. Odd terms like “warm power age”, “creating nations”, and “electrochemical statement” suggest that machine-translator has been used. Sentences like “An extraordinary power (25 kV) was useful to the gathering objective drum by a high voltage power source.” are totally incomprehensible.
  2. The novelty of this study remains unclear. Is there any new methodology? Why was HEMA selected as the functional group?

Detailed comments:

  1. The manuscript must be first rewritten and only then any detailed comments are possible.

Author Response

  1. The language must be re-checked. Odd terms like “warm power age”, “creating nations”, and “electrochemical statement” suggest that machine-translator has been used. Sentences like “An extraordinary power (25 kV) was useful to the gathering objective drum by a high voltage power source.” are totally incomprehensible.

Response 1: the odd terms that you are mentioned are revised

  1. The novelty of this study remains unclear. Is there any new methodology? Why was HEMA selected as the functional group?

Response 1: Functional monomers should be rationally chosen according to the nature of the template ions, coordination number, coordination force, and the interaction between the functional monomer and the template ions.

Round 2

Reviewer 2 Report

There are hundreds of publications describing the performance of different natural or synthetic materials for adsorbing metals in solution. From this point of view, the present manuscript is not very original. On the other hand, the production of adsorbent based on cellulose nanofibers is of some interest to specialists in the field. The manuscript is also well written and easy to understand. I therefore recommend the publication of this manuscript.

Author Response

thank you for your kind reviewing of my manuscript and I would like to inform you that, besides your important notice to us, we sent the manuscript to the language editing center to be sure from the English of the manuscript. 

Reviewer 3 Report

The authors have made only cosmetic changes and these do not improve the quality of the manuscript. My recommendation was to rewrite the paper in understandable English. This has not been done and the manuscript should be rejected.

Author Response

My recommendation was to rewrite the paper in understandable English. 

response: Herewith the attached file regarding the MPDI language edition center certified that the manuscript has been re-edited.
